# The Relationship Between Premenstrual Dysphoric Symptoms, Perceived Stress, and Sleep Quality Among Nursing Students: A Cross-Sectional Study

**DOI:** 10.3390/healthcare13222862

**Published:** 2025-11-11

**Authors:** Adnan Innab, Atallah Alenezi, Asmaa Khaled, Mashael Dewan, Mona Mostafa

**Affiliations:** 1Nursing Administration and Education Department, College of Nursing, King Saud University, Riyadh 11421, Saudi Arabia; 2College of Nursing, Imam Mohammad Ibn Saud Islamic University (IMSIU), Riyadh 11623, Saudi Arabia; akhalenezi@imamu.edu.sa; 3Nursing Sciences Department, College of Applied Medical Sciences, Shaqra University, Shaqra 11961, Saudi Arabia; khaled@su.edu.sa; 4Community Health Nursing, Faculty of Nursing, Alexandria University, Alexandria 21526, Egypt; 5Adult Health Nursing Department, Dr. Suliman Al Habib College for Knowledge, Riyadh 12451, Saudi Arabia; mashael.dewan@drsulaimanalhabib.edu.sa; 6Department of Nursing Sciences, College of Applied Medical Sciences, Shaqra University, Shaqra 11961, Saudi Arabia; m.mohammad@su.edu.sa; 7Psychiatric & Mental Health Nursing Department, Faculty of Nursing, Cairo University, Cairo 12613, Egypt

**Keywords:** premenstrual dysphoric mood disorder, stress, sleep disturbance, sleep quality

## Abstract

Premenstrual dysphoric disorder (PMDD) is a psychological condition that significantly influences female students’ behavior, cognitive ability, and mental health status. It has a critical role in mental health and well-being and in the academic success and professional performance of nursing students. Objectives: This study aimed to assess the relationship between PMDD, perceived stress levels and sleep quality among female nursing students in Saudi Arabia. Methods: This study utilized a cross-sectional, correlational, descriptive design. The sample size was 144 students, and we used a convenience sampling technique. Data were collected using the following tools: Demographics and Gynecological Data Sheet, Premenstrual Symptoms Screening Tool, Pittsburgh Sleep Quality Index, and Perceived Stress Scale. Results: The levels of premenstrual symptoms and sleep disturbances among the participants can be regarded as moderate, whereas the level of perceived stress was found to be at its highest level, with 63.9% of participants experiencing a moderate level and 14.6% reporting severe stress. There was a significant positive correlation between premenstrual symptoms and perceived stress (r = 0.39, *p* = 0.005). Thus, as premenstrual symptoms increase, so does perceived stress. Additionally, there is a noteworthy correlation between perceived stress and sleep disturbance (r = 0.28, *p* = 0.04), indicating that higher levels of stress are associated with more sleep disturbances. Conclusions: PMDD symptoms have a detrimental influence on female students’ emotional state; thus, mental health experts play an important role in identifying variables that mitigate the severity of PMDD among female’s students.

## 1. Introduction

Premenstrual dysphoric mood disorder (PMDD) is a complex condition caused by the fluctuations of sex steroid hormones associated with the menstrual cycle. Symptoms manifest approximately one week before menstruation, and include irritability, anxiety, mood instability and various physical complaints such as headache, breast tenderness and bloating [1]. While the exact etiology of PMDD remains unclear, contributing factors include genetics, sociocultural influences, central nervous system sensitivity to reproductive hormones, and abnormalities in brain neurotransmitters, particularly serotonergic dysfunction [2,3]. PMDD is characterized by mood disturbances and physical symptoms that resolve with the onset of menstruation. Additionally, a history of mental health disorders increases susceptibility to PMDD [4,5].

Epidemiological studies indicate that most young women experience premenstrual symptoms during their reproductive years, with severe symptoms classified as PMDD [6]. Approximately 80% of women in their reproductive years report experiencing physical, emotional, or behavioral symptoms before menstruation [7]. The Diagnostic and Statistical Manual of Mental Disorders, Fifth Edition (DSM-5), estimates PMDD prevalence to range from 1.8% to 5.8% annually, with higher rates observed in certain populations [8].

### 1.1. Perceived Stress and PMDD

Perceived stress is an individual’s subjective evaluation of the stress experienced over a specific period [9]. Research has found that perceived stress among nursing students negatively impacts their clinical performance [10,11], academic achievement [12], and patient care [13]. PMDD is another challenging factor that negatively affects female nursing students’ health due to pain, sleep disturbance and hormonal change [14]. Research indicates a complex relationship between stress and PMDD, with stress acting as both a risk factor and consequence of premenstrual symptoms. For instance, stress exacerbates the severity of PMDD and is linked to increased symptom persistence [11]. University students frequently report stress and menstrual difficulties, significantly impacting their academic performance and quality of life [15,16,17]. Another study by Al-Shahrani et al. (2021) found that PMS had a substantial detrimental impact on their everyday activities [18]. Similarly, Nisar and colleagues discovered that premenstrual syndrome impairs students’ academic performance and well-being [16]. These studies have been influential in shedding light on women with PMDD. However, the impact of PMDD on perceived stress among nursing students in Saudi Arabia remains underexplored, with limited information on how it may affect their mental health.

### 1.2. PMDD and Sleep Quality

PMDD is linked with severe functional impairment in quality of life and functional capacity [19,20]. According to studies, women with PMS or PMDD complain of sleep-related issues such as difficulties sleeping, exhaustion, and poor concentration [21]. Furthermore, sleep disorders such as insomnia or hypersomnia are one of the DSM-IV defining criteria [19,22], and they affect around 70% of women with PMDD [23]. A study by Parry and his colleagues found no difference in sleep EEG variables between controls and study group [24], other studies showed significantly increase rapid eye movement (REM) sleep onset latency [20], increase stage-two sleep or reduce REM sleep when compared to healthy controls. Additionally, research has found that PMDD can affect the sleep quality of nursing students [14,25]. Therefore, identifying key predictors could help guide the development of effective therapies and enhance understanding of PMDD among students early in their reproductive years, allowing for earlier intervention.

The purpose of this study was to assess the relationship between PMDD, perceived stress levels and sleep quality among female nursing students in Saudi Arabia. The specific aims of this study were the following: (1) to determine the prevalence of PMDD, perceived stress, and sleep quality among nursing students; and (2) to assess the relationship between demographic characteristics and premenstrual symptoms, perceived stress, and sleep quality. We hypothesized that high levels of PMDD would be associated with high levels of perceived stress and low levels of sleep quality. In this study, premenstrual dysphoric disorder (PMDD) was operationally defined as the occurrence of at least five symptoms (including at least one mood-related symptoms such as irritability, depressed mood, anxiety, or affective lability) during the luteal phase of the menstrual cycle (i.e., the week before menstruation). Symptoms must cause clinically significant distress or functional impairment and improve within a few days after the onset of menstruation.

## 2. Materials and Methods

### 2.1. Design

This was a cross-sectional, correlational, descriptive study. This study followed the Strengthening the Reporting of Observational Studies in Epidemiology (STROBE) guidelines.

### 2.2. Sample and Setting

Data was collected using a convenience sampling technique. Participants were recruited from a governmental university in Riyadh, Saudi Arabia. Nursing students were chosen as the study population due to their unique combination of academic and clinical demands. They experience a high academic workload alongside exposure to clinical environments, which can contribute to elevated stress levels. These factors may influence both the severity and perception of premenstrual symptoms, making nursing students a particularly relevant and informative group for this study. The inclusion criteria comprised female undergraduate nursing students enrolled in the nursing department who voluntarily agreed to participate in the study. Students were excluded if they had been diagnosed with psychiatric or gynecological disorders or if they were regularly taking hormonal or psychotropic medications.

The sample size was determined using G*Power 3.1.9 software (Heinrich Heine University, Düsseldorf, Germany). The parameters for conducting the inferential statistics included an alpha level of 0.05, a power of 0.8, and an effect size of 0.15. To account for potential missing data, we decided to increase the sample size by 15%, resulting in a total of 140 participants. The total sample size of this study was 144.

### 2.3. Measurements

Demographic and gynecological data include code, age, level of education, marital status, age of menarche, Body Mass Index, drug use, period length, regularity of period, knowledge of premenstrual symptoms, family history of dysmenorrhea, and gynecological problems. The Premenstrual Symptoms Screening Tool (PSST) was developed by Steiner et al. [6], and it helps clinicians to identify and assess the severity of premenstrual syndrome symptoms. It consists of 19 items that cover various symptoms that are commonly associated with PMDD, including mood swings, irritability, anxiety, sadness, and fatigue. The tool measuring severity of symptoms using a four-point scale. The tool demonstrates excellent reliability (α = 0.94).

The English version of the Pittsburgh Sleep Quality Index (PSQI) developed by Buysse et al. (1989) was used in this study [26]. The PSQI is a 19-item scale that assesses various components of sleep quality, including subjective sleep quality, sleep latency (the time it takes to fall asleep), sleep duration, sleep efficiency, sleep disturbances, use of sleeping medications, and daytime dysfunction. The greater the scores, the poorer sleep quality. In previous study in Saudi Arabia, the PSQI has demonstrated good psychometric properties and had good Cronbach’s alpha of 0.9 [27]. In the current study, the tool demonstrates good reliability (α = 0.79).

The perceived stress scale (PSS) developed by Sheldon Cohen and colleagues and its psychometric properties were tested in Taylor’s study [28]. It assesses the level of perceived stress over the past month, focusing on feelings of unpredictability, uncontrollability, and overload. It consists of 10-item measuring psychological stress levels on a 5-point Likert scale, ranging from 0 (never) to 4 (very often). Among university students in Saudi Arabia, the scale showed acceptable reliability, with a Cronbach’s alpha of 0.78 [29]. In this study, the scale has demonstrated good reliability, with a Cronbach’s α of 0.86.

### 2.4. Data Collection

The current study utilizes a web-based survey platform (Google Form) to collect data. After obtaining formal approval from relevant authorities, a survey was created with Google Forms. To avoid missing data, all questions were mandatory, and participants were not allowed to submit multiple responses from the same account. The survey link was sent to students via social media (WhatsApp groups and Twitter). Data collection was performed from March to June 2024; then, the responses were downloaded and analyzed.

### 2.5. Data Analysis

Statistical analyses were performed using SPSS Version 30.0. Descriptive statistics (i.e., frequency, percentages, mean, and standard deviations) were used to describe the demographic characteristics and main variables (premenstrual symptoms, perceived stress, and sleep disturbance). Pearson’s product moment correlation was used to determine the association between premenstrual symptoms, perceived stress and sleep disturbance.

### 2.6. Ethical Considerations

Approval was obtained from Shaqra University’s Institutional Review Board (IRB) under reference number [ERC_SU_S_202400002, approval date: 19 February 2024]. This study adhered to the principles of the Declaration of Helsinki. Informed consent was obtained from all participants prior to data collection, who were made aware of the study’s purpose, along with the associated risks and benefits. Their privacy was ensured by not collecting identifiers and reporting data in aggregate form. Prior to starting the survey, informed consent was obtained from all participants who agreed to take part. To further protect their privacy, all collected data was anonymized and treated as confidential.

## 3. Results

### 3.1. Sample Characteristics

The demographic characteristics of the participants are presented in Table 1. The majority of students (57.7%) were aged between 19 and 21 years and were in their second year of nursing (48.6%). Additionally, 97.2% were single, and 50% had a normal BMI. The average age of menarche was 13.2 years (SD ± 3.0). Furthermore, the table shows that 83.3% had regular periods, 90.3% were aware of premenstrual symptoms, and 51.4% reported no family history of dysmenorrhea.

### 3.2. Premenstrual Symptoms, Perceived Stress and Sleep Quality

The results indicated, as shown in Table 2, that the average score for premenstrual symptoms was 23.7 out of a possible 56, accounting for 42.4%. The mean perceived stress score was 18.9 out of a maximum of 31, representing 61%, while the average score for sleep disturbances was 8.3 out of 17, equating to 48.9%. This suggests that the levels of premenstrual symptoms and sleep disturbances among the participants can be regarded as moderate, whereas the level of perceived stress is considered above average.

Additionally, as shown in Table 3, perceived stress was found to be at its highest level, with 63.9% of participants experiencing a moderate level and 14.6% reporting severe stress. In contrast, 61.1% of the sample had mild sleep disturbances, while 68.1% experienced mild premenstrual symptoms. Only 4.2% reported severe premenstrual symptoms.

### 3.3. Correlations Between Premenstrual Symptoms, Perceived Stress, and Sleep Disturbance

The results presented in Table 4 reveal significant associations between the study variables. A positive correlation was observed between premenstrual symptoms and perceived stress (r = 0.39, *p* = 0.005), indicating that higher levels of premenstrual symptoms are associated with greater perceived stress. Furthermore, perceived stress demonstrated a significant positive correlation with sleep disturbance (r = 0.28, *p* = 0.04), suggesting that increased stress levels are linked to poorer sleep quality.

### 3.4. Correlations Between Demographic Data of Studied Sample and Study Variables

Table 5 presents the correlation between demographic characteristics and premenstrual symptoms, perceived stress, and sleep. Despite the relatively small magnitude of most significant correlations, a number of meaningful associations were observed. A significant inverse correlation was observed between premenstrual symptoms and both age and education level, suggesting that the severity of these symptoms tends to decrease with increasing age and educational attainment. In contrast, premenstrual symptoms showed a significant positive correlation with both symptom awareness and menstrual regularity, indicating that greater awareness and regular cycles may be associated with more pronounced symptom experiences. Moreover, an inverse correlation was identified between age and perceived stress, implying that older individuals may report lower levels of stress. These findings emphasize the complex interplay between demographic factors and women’s health experiences.

## 4. Discussion

PMDD is a severe form of premenstrual syndrome that has increasingly captured global attention [30]. It profoundly impacts the academic performance and overall quality of life of female students [31]. The onset of menarche, occurring during adolescence, represents a significant biological transition from childhood to adulthood. Beyond being an indicator of a woman’s biological development, the age at which menarche occurs is also considered a crucial metric for assessing a girl’s quality of life [32]. Understanding these dynamics highlights the importance of addressing PMDD and related menstrual issues to support young women’s health and well-being.

A body of evidence indicated that a secular decline in the age of menarche onset over the past few decades [33]. This evidence is in line with the current study results, which stated that more than half of respondents had menarches between the ages of 11 and 15, with a mean of 13.2 ± 3.0 years, and most of them had regular periods. These findings align well with the research conducted by Anikwe et al. (2020), which reported a mean age of menarche at 13 years (SD ± 1.0) among adolescent girls, with more than half experiencing regular menstrual cycles [31]. Similarly, a study from Lebanon indicated that most participants began menstruating between the ages of 12 and 14, with a majority also reporting regular monthly periods [35]. Furthermore, additional supporting evidence from Marques et al. (2022) revealed that the average age of menarche among teenagers was 12.4 years, with most experiencing a menstrual flow that lasts less than six days, and over half maintaining a regular menstrual cycle [36]. Collectively, these studies highlight a consistent trend in the age of menarche and menstrual regularity among adolescent girls across different regions. Important to note, early menarche among young females is associated with high stress levels and sleep disturbance due to hormonal shift and pain [37,38]. Therefore, future interventional studies that examine the management of PMDD are recommended [39].

It is noteworthy that a significant majority of young female students demonstrated an awareness of premenstrual symptoms, as indicated by the current study’s results. This finding aligns closely with research conducted in Karachi, Pakistan, which surveyed 448 female students and found a similarly high level of awareness regarding PMS [40]. This heightened awareness among the participants may be attributed to their educational backgrounds in the medical field, which likely provided them with valuable knowledge about premenstrual symptoms. Such insights underscore the importance of education in enhancing understanding of health issues among young women.

Premenstrual symptoms are a prevalent disorder that affects young female adolescents’ physical, behavioral, and mental health. It is linked to strained social and familial ties, absenteeism and interference at work, and higher medical expenses [41]. When comparing the findings of the present study to other research, it is evident that the severity of premenstrual symptoms and sleep disturbances in the studied sample can be classified as moderate, while the level of perceived stress is notably above average. Specifically, the mean score for premenstrual symptoms was 23.7 out of a possible 56, which reflects over two-fifths of the maximum. The mean perceived stress score stood at 18.9 out of 31, indicating that more than half of the participants experienced significant stress. Additionally, the mean score for sleep disturbances was 8.3 out of 17, representing slightly less than half of the female students. This aligns with findings from Yi et al. (2023), which reported a range of premenstrual symptom severity from normal to premenstrual dysphoric disorder (PMDD) among 143 university students in South Korea [42]. These insights emphasize the need for awareness and support regarding menstrual health and stress management among young women.

Furthermore, it can be inferred that the intensity of monthly pain serves as a critical indicator of the severity of premenstrual syndrome (PMS), which is a complex disorder characterized by emotional, behavioral, and physical symptoms. According to estimates from the World Health Organization, approximately 20–31% of female university students worldwide experience at least one mental disorder related to menstruation. This statistic underscores the widespread impact of PMS on women’s mental health and highlights the importance of addressing these issues in academic and healthcare settings. Recognizing the interconnectedness of menstrual health and overall well-being is crucial for providing effective support to young women [43]. This suggests that pain is associated with several PMS symptoms. In the same study, it was found that the majority of participants experienced a moderate level of stress, with over one-tenth of the sample reporting severe stress levels. Additionally, more than half of the participants exhibited mild levels of both sleep disturbances and premenstrual symptoms. These findings highlight the varying degrees of psychological and physical challenges faced by the participants, emphasizing the importance of targeted interventions to improve mental health and overall well-being among young women.

Consequently, university female students’ relationships, daily routines, quality of life, and academic performance may all be negatively impacted by the severity of their PMS [18]. Consistently, a study conducted in Koç University in Istanbul revealed that over half of the participants reported dysmenorrhea, which may have an impact on their academic performance, primarily because of absenteeism [31].

It is noteworthy that a positive correlation exists between premenstrual symptoms and perceived stress, as well as between perceived stress and sleep disturbances. However, there is no significant relationship between premenstrual symptoms and sleep disturbances. These findings closely align with a recent study conducted by Alshdaifat (2022), which demonstrated a significant correlation between the severity of premenstrual symptoms, perceived stress, and dysmenorrhea [44]. This connection underscores the complex interplay between these factors, highlighting the need for further examination and understanding of how they influence women’s health and well-being [44].

Several factors may influence premenstrual symptoms among female students, including age, education level, age at menarche, menstrual regularity and length, family history of dysmenorrhea, and knowledge about premenstrual symptoms. The findings of the current study indicate a significant inverse correlation between premenstrual symptoms and both age and education level, suggesting that as age and education increase, the severity of premenstrual symptoms may decrease. Conversely, a positive correlation was observed between premenstrual symptoms and knowledge of these symptoms, implying that greater awareness and understanding may be linked to increased symptom reporting. These insights highlight the importance of education and awareness in managing premenstrual symptoms among young women.

Additionally, one of the main factors affecting sleep throughout the menstrual cycle is the dynamic fluctuation of female hormones, notably estrogen and progesterone. These hormones experience major variations as the menstrual cycle goes on, and these changes may affect sleep patterns during the luteal phase, when progesterone levels rise after ovulation. Because this rise in progesterone is linked to elevated body temperature, this can make it challenging for some women, who may find it more difficult to get a good night’s sleep.

The findings of the present study pointed out that a positive correlation exists between sleep and the regularity of periods. Similarly, the study of Rugvedh et al. (2023) highlights the effect of sleep disturbances on menstrual disturbances, including PMS, dysmenorrhea, an abnormal menstrual cycle, and heavy bleeding during periods [45]. Jeon and Baek (2023) provide more support by elucidating the relationship among the menstrual cycle, sleep duration, and cardiovascular health [46]. Despite having a medical background, only 19% of young women were aware of premenstrual dysphoric disorder [32]. This further demonstrates that there is still a lack of awareness about prevalent health issues affecting women.

### 4.1. Study Limitations

While this study provides valuable insights into the factors influencing premenstrual symptoms, its cross-sectional design restricts the ability to establish causal relationships. To deepen understanding in this field, future research should consider employing longitudinal designs, which allow for the observation of changes over time and the exploration of causal links. Involving a diverse range of populations will also enhance the generalizability of the findings, making them more applicable to a broader audience. Additionally, it is important to note that the use of self-reported measures may introduce biases related to social desirability, as participants may be inclined to present their experiences in a more favorable light. Recall bias is another factor that may affect the current study’s findings. Participants may wrongly report the previous events, or underreport or overreport the symptoms [47]. Thus, future research is recommended to shorten the period between the event and data collection. Addressing these limitations in future studies could lead to more accurate and reliable insights into the complexities of premenstrual symptoms and their various influencing factors.

### 4.2. Implications for Practice

Raising awareness of PMDD and its associated factors can inform health promotion strategies and health policies for university students to support female students and increase awareness on this issue. Early interventions targeting stress reduction and sleep quality, such as virtual-reality-based meditation may improve students’ sleep quality and stress levels [37].

## 5. Conclusions

Premenstrual dysphoric disorder (PMDD) significantly affects the mood states of female students, highlighting the need for increased awareness and intervention. Enhancing self-awareness along with implementing psychological and psychiatric support can help students manage PMDD, reduce stress levels, and improve their overall quality of life. Furthermore, additional research is essential to explore the associations between sleep problems and long-term outcomes in students experiencing PMDD. Future longitudinal design research is recommended to confirm the direction of causality between stress and sleep. Addressing these issues will contribute to a better understanding of PMDD and the comprehensive support required for affected individuals.

## Figures and Tables

**Table 1 healthcare-13-02862-t001:** Demographic data of the studied students (n = 144).

Demographic Data	Frequency	Percentage	M ± SD
Students age in years			
19–21	83	57.7	21.1 ± 1.3
22–24	61	42.3	
Level of education			
Level 2	39	27.1	
Level 3	35	24.3	
Level 4	70	48.6	
Students’ marital status			
Single	142	98.6	
Married	2	1.4	
Body Mass Index			
fUnderweight 18.5	57	39.6	
Normal 25.0–29.9	72	50.0	
Overweight	13	9.0	
Obese	2	1.4	
Age of Menarche			
5–10	15	10.4	13.2 ± 3.0
11–15	99	68.8	
>15	30	20.8	
Period length (days)			
1–5	29	20.1	
6–10	115	79.9	
Regularity of period			
Irregular	24	16.7	
Regular	120	83.3	
Knowledge of premenstrual symptoms			
No	14	9.7	
Yes	130	90.3	
Family history of dysmenorrhea			
No	74	51.4	
Unknown	51	35.4	
Yes	19	13.2	
Gynecological problems			
Endometriosis	7	4.8	
Fibro	11	7.6	
PCO	55	38.1	
PMDD	33	22.9	
Vaginal infection	38	26.3	

Note: Polycystic Ovary (PCO).

**Table 2 healthcare-13-02862-t002:** Mean score of the study variables (n = 144).

Study Variables	Min	Max	Mean ± SD	Percent
Premenstrual symptoms	0.0	56.0	(23.7 ± 11.11)	42.4
Perceived stress	6.0	31.0	(18.90 ± 4.57)	61.0
Sleep disturbance	0.0	17.0	(8.31 ± 3.69)	48.9

**Table 3 healthcare-13-02862-t003:** Levels of premenstrual symptoms, perceived stress and sleep disturbance among the studied sample (n = 144).

Levels	Premenstrual Symptoms		Perceived Stress		Sleep Disturbance	
	*f*	%	*f*	%	*f*	%
Mild	98	68.1	31	21.5	88	61.1
Moderate	40	27.8	92	63.9	43	29.9
Severe	6	4.2	21	14.6	13	9.0

**Table 4 healthcare-13-02862-t004:** Correlation matrix for menstrual symptoms, perceived stress, and sleep disturbance among the studied students (n = 144).

	1	2	3
1. Premenstrual symptoms	1		
2. Perceived stress	0.39 **	1	
3. Sleep disturbance	0.17	0.28 *	1

* *p* < 0.05, ** *p* < 0.01.

**Table 5 healthcare-13-02862-t005:** Correlation between demographic data and studied variables among the studied students (n = 144).

Demographic Data	Premenstrual Symptoms	Perceived Stress	Sleep
	R	R	R
Age	−0.24 *	−0.16 *	−0.07
Levels of education	−0.21 *	−0.10	0.05
BMI	−0.01	−0.04	−0.1
Menarche age	0.004	−0.02	0.04
Period length (days)	−0.12 *	−0.03	−0.13
Regularity of period	0.03	0.06	0.16 *
Knowledge of premenstrual symptoms	0.22 *	−0.03	0.12
Family history of dysmenorrhea	−0.18 *	0.09	0.09

* *p* < 0.05.

## Data Availability

The datasets are available from the corresponding author upon reasonable request.

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
