# Peer review of "The Relationship Between Premenstrual Dysphoric Symptoms, Perceived Stress, and Sleep Quality Among Nursing Students: A Cross-Sectional Study"

_healthcare, 2025, doi:10.3390/healthcare13222862_

Round 1

Reviewer 1 Report

Comments and Suggestions for Authors

After carefully reviewing the manuscript, I believe that it addresses an important and relevant topic. However, there are several substantial methodological and reporting issues that need to be resolved before the paper can be considered for publication.

The study’s aim is not sufficiently justified. It should be clarified in the Introduction section why the research was conducted in Saudi Arabia and specifically among nursing students.

I recommend that the authors explicitly state the main hypotheses of the study in the Introduction.

In the Methods section, the inclusion and exclusion criteria for participants are not specified. This is a major limitation in terms of sample homogeneity and the reliability of the findings. Factors such as participants’ health status, existing psychiatric or gynecological conditions, and regular medication use can directly affect PMDD, stress, and sleep quality. Therefore, I strongly recommend that the authors clearly define these criteria.

(Page 3, line 95) The authors reported an effect size of 15 in the power analysis. Could this be a typographical error? Standard effect sizes such as Cohen’s d, f², or r are generally classified as 0.1 (small), 0.3 (medium), and 0.5 (large).

It is unclear whether the measurement instruments used in the study have been validated and tested for reliability in Saudi Arabia. Which versions of the scales were used? If cultural adaptations exist and were applied, the corresponding studies should be cited.

In the Results section, most of the reported correlation coefficients are in the low-to-moderate range. I find that these relationships are sometimes presented in an exaggerated manner.

Although the correlations were statistically significant, their low-to-moderate effect sizes were not discussed. To better reflect the clinical and practical significance of the findings, this issue should be addressed in the Discussion.

In several places in the manuscript (e.g., line 18, line 247, line 321), the term Premenstrual Dysphoric Disorder is written in full. After introducing the abbreviation (PMDD) at first mention, the manuscript should consistently use the abbreviation throughout.

Author Response

Comment 1: 

The study’s aim is not sufficiently justified. It should be clarified in the Introduction section why the research was conducted in Saudi Arabia and specifically among nursing students.

Response 1: 

Thank you for your feedback. The edits have been made as requested and integrated in the introduction section (P2-3). 

Comment 2: I recommend that the authors explicitly state the main hypotheses of the study in the Introduction.

Response 2: Thank you for your feedback. We included the hypothesis after study aims (p3). 

Comment 3: The inclusion and exclusion criteria for participants are not specified. This is a major limitation in terms of sample homogeneity and the reliability of the findings.

Response 3: 

We appreciate the reviewer’s valuable comment and acknowledge the importance of clearly defining inclusion and exclusion criteria to ensure sample homogeneity and the reliability of findings. In the revised version, the inclusion criteria have now been explicitlystated as follows:

“The inclusion criteria comprised female undergraduate nursing students enrolled in the nursing department who voluntarily agreed to participate in the study. Students were excluded if they had been diagnosed with psychiatric or gynecological disorders or if they were regularly taking hormonal or psychotropic medications.”

This clarification has been added to the Methods section (page 3, Line 108-117) to strengthen the methodological rigor of the study.

Comment 4: Factors such as participants’ health status, existing psychiatric or gynecological conditions, and regular medication use can directly affect PMDD, stress, and sleep quality. Therefore, I strongly recommend that the authors clearly define these criteria. 

Response 4: We sincerely thank the reviewer for this valuable comment. We agree that factors such as participants’ health status, existing psychiatric or gynecological conditions, and regular medication use can influence PMDD, stress, and sleep quality. In response to this suggestion, the exclusion criteria have been updated to clearly state that students diagnosed with psychiatric or gynecological disorders, as well as those regularly taking hormonal or psychotropic medications, were excluded from the study. This clarification has been added to the Methods section (page 3) to enhance the accuracy and validity of the result.

Comment 5: (Page 3, line 95) The authors reported an effect size of 15 in the power analysis. Could this be a typographical error? Standard effect sizes such as Cohen’s d, f², or r are generally classified as 0.1 (small), 0.3 (medium), and 0.5 (large).

Response 5: 

We thank the reviewer for this important observation. We acknowledge that the reported effect size of 15 was a typographical error. The correct value used in the power analysis was 0.15, representing a small-to-medium effect size, in accordance with Cohen’s conventions. This correction has been made in the Methods section (page 3,of the revised manuscript). 

Comment 6: It is unclear whether the measurement instruments used in the study have been validated and tested for reliability in Saudi Arabia. Which versions of the scales were used? If cultural adaptations exist and were applied, the corresponding studies should be cited.

Response 6: 

Thanks for your valuable comment. The instruments were not translated into Arabic.

PSQI:

“The English version of the Pittsburgh Sleep Quality Index (PSQI) developed by Buysse et al. (1989) was used in this study [26].The PSQI is a 19-item scale that assesses various components of sleep quality, including subjective sleep quality, sleep latency (the time it takes to fall asleep), sleep duration, sleep efficiency, sleep disturbances, use of sleeping medications, and daytime dysfunction. The greater the scores, the poorer sleep quality. In previous study in Saudi Arabia, the PSQI has demonstrated good psychometric properties and had good Cronbach's alpha of 0.9 [27]. In the current study, the tool demonstrates good reliability (α = 0.79).”

PSS:

“The Perceived stress scale (PSS) by Sheldon Cohen and colleagues and its psychometric properties were tested in Taylor’s study[28]. It assesses the level of perceived stress over the past month, focusing on feelings of unpredictability, uncontrollability, and overload. It consists of 10-item measuring psychological stress levels on a five-point Likert scale, ranging from 0 (never) to 4 (very often). Among university students in Saudi Arabia, the scale showed acceptable reliability, with a Cronbach’s alpha of 0.78 [29].  In this study, the scale has demonstrated good reliability with Cronbach’s α of 0.86.”

Comment 7: In the Results section, most of the reported correlation coefficients are in the low-to-moderate range. I find that these relationships are sometimes presented in an exaggerated manner. 

Response 7: Thanks for your feedback. To address this concern, we have revised the manuscript to clarify that the observed correlations indicate associations rather than strong predictive relationships, and we have moderated the language in the Results and Discussion sections to reflect the modest strength of these correlations.

Comment 8: In several places in the manuscript (e.g., line 18, line 247, line 321), the term Premenstrual Dysphoric Disorder is written in full. After introducing the abbreviation (PMDD) at first mention, the manuscript should consistently use the abbreviation throughout.

Response 8: We totally agree with you. The revised version of the manuscript included only the abbreviation of PMDD.  

Reviewer 2 Report

Comments and Suggestions for Authors
  1. Methodological rigour and definition of variables (PMDD)

The most important point is clarity regarding the main variable (PMDD):

  • Clarify the diagnosis/screening (PMDD): The title and content refer to premenstrual dysphoric disorder (PMDD), a clinical diagnosis according to the DSM-5. It is crucial that the authors explain unequivocally whether they used a screening tool (e.g., PMDD-SR or similar) or strictly applied the DSM-5 diagnostic criteria. If a screening tool was used, the title should more accurately reflect that the symptoms of PMDD/severe premenstrual syndrome are being studied, not necessarily the clinical prevalence of the disorder.
  • Justification of samples: The article mentions a considerable sample size, which is a strength. However, the choice of nursing students as the population should be further justified, highlighting their high academic workload, exposure to clinical stress, and how this makes them a particularly interesting group for the study of these variables.
  • Psychometric properties: It is recommended to add a sentence in the Methods section confirming the validity and reliability (e.g., Cronbach's alpha value) of the scales used (Perceived Stress Scale, Pittsburgh Sleep Quality Index) in the study population (nursing students in Saudi Arabia), or at least cite their previous use in similar populations.
  1. Limitations of cross-sectional design

The study design is cross-sectional, which is an inherent limitation that should be emphasised, especially in the Discussion section:

  • Avoid inferences of causality: The text should be very careful when discussing the results. A cross-sectional study can only establish associations or correlations between TDPM, stress, and sleep quality. Any language suggesting that ‘stress causes poor sleep quality’ or vice versa should be avoided. It should be changed to phrases such as ‘stress is significantly associated with...’.
  • Recall bias: In the Limitations section, it is essential to add recall bias as a limitation, given that all data were collected through self-report (questionnaires) and participants must recall past symptoms and habits.
  1. Statistical rigour and discussion of results
  • Regression model: In the results section, when presenting the linear regression analysis, it must be confirmed that the model assumptions were met (such as normality of residuals and absence of multicollinearity).
  • R2 value: It is important to highlight and discuss the adjusted R2 value of the regression models. This value indicates the percentage of variance in sleep quality (or stress) that is explained by the other variables in the model. A low R2 would reduce the impact or ‘practical significance’ of the findings.
  • Relevance of the findings: The discussion is the ideal place to delve deeper into why the relationship found is important. For example, if TDPM is found to be a stronger predictor of poor sleep quality than perceived stress, this should be the main focus of the argument.
  1. Conclusion and Projections
  • Specific Recommendations: Recommendations in the conclusion and discussion should be more actionable. Instead of simply stating that more research is needed, the following suggestions could be made:
  • Recommend longitudinal studies to confirm the direction of causality between stress and sleep.
  • Suggest the implementation of specific stress management interventions (e.g., mindfulness or cognitive therapy) in the nursing curriculum to measure their impact on RMD symptoms and sleep quality.

Author Response

Comment 1: Methodological rigor and definition of variables (PMDD). The most important point is clarity regarding the main variable (PMDD).

Response 1: We appreciate the reviewer’s observation regarding the clarity of the main variable (PMDD). We would like to clarify that PMDD was conceptually defined in the First paragraph of the Introduction section in the manuscript, where the theoretical background of the disorder was discussed in detail. However, to strengthen the study and ensure clarity in the operational definition (Page 2). This addition ensures consistency between the theoretical and methodological sections and enhances the transparency of variable measurement.

Comment 2: Clarify the diagnosis/screening (PMDD): The title and content refer to premenstrual dysphoric disorder (PMDD), a clinical diagnosis according to the DSM-5. It is crucial that the authors explain unequivocally whether they used a screening tool (e.g., PMDD-SR or similar) or strictly applied the DSM-5 diagnostic criteria. If a screening tool was used, the title should more accurately reflect that the symptoms of PMDD/severe premenstrual syndrome are being studied, not necessarily the clinical prevalence of the disorder.

Response 2: We thank the reviewer for this important observation. In our study, we used the Premenstrual Symptoms Screening Tool (PSST) developed by Steiner et al. This 19-item self-report questionnaire identifies the presence and severity of symptoms commonly associated with PMDD, including mood, behavioral, and physical symptoms. It is a screening tool and does not provide a formal clinical diagnosis according to DSM-5 criteria. Accordingly, we have revised the title to better reflect the nature of our study: The Relationship Between Premenstrual Dysphoric Symptoms, Perceived Stress, and Sleep Quality Among Nursing Students: A Cross-Sectional Study.

Additionally, we have clarified in the Methods section that the PSST was used to screen and quantify the severity of premenstrual symptoms, rather than to diagnose PMDD clinically. These revisions ensure that the manuscript accurately represents the study design and the scope of our findings.

Comment 3: The article mentions a considerable sample size, which is a strength. However, the choice of nursing students as the population should be further justified, highlighting their high academic workload, exposure to clinical stress, and how this makes them a particularly interesting group for the study of these variables.

Response 3: 

We sincerely thank the reviewer for drawing our attention to this valuable addition which allowed us to provide a clearer justification for selecting nursing students as a target population for this study. Nursing students were chosen as the study population due to their unique combination of academic and clinical demands. They experience a high academic workload alongside exposure to clinical environments, which can contribute to elevated stress levels. These factors may influence both the severity and perception of premenstrual symptoms, making nursing students a particularly relevant and informative group for examining the relationship between stress, sleep quality, and premenstrual symptom. This justification was highlighted in the main manuscript (page 3).

Comment 4: It is recommended to add a sentence in the Methods section confirming the validity and reliability (e.g., Cronbach's alpha value) of the scales used (Perceived Stress Scale, Pittsburgh Sleep Quality Index) in the study population (nursing students in Saudi Arabia), or at least cite their previous use in similar populations.

Response 4: Thanks for your feedback. 

The English version of the Pittsburgh Sleep Quality Index (PSQI) developed by Buysse et al. (1989) was used in this study [26]. The PSQI is a 19-item scale that assesses various components of sleep quality, including subjective sleep quality, sleep latency (the time it takes to fall asleep), sleep duration, sleep efficiency, sleep disturbances, use of sleeping medications, and daytime dysfunction. The greater the scores, the poorer sleep quality. In previous study in Saudi Arabia, the PSQI has demonstrated good psychometric properties and had good Cronbach's alpha of 0.9 [27]. In the current study, the tool demonstrates good reliability (α = 0.79).

PSS:

The Perceived stress scale (PSS) by Sheldon Cohen and colleagues and its psychometric properties were tested in Taylor’s study[28]. It assesses the level of perceived stress over the past month, focusing on feelings of unpredictability, uncontrollability, and overload. It consists of 10-item measuring psychological stress levels on a five-point Likert scale, ranging from 0 (never) to 4 (very often). Among university students in Saudi Arabia, the scale showed acceptable reliability, with a Cronbach’s alpha of 0.78 [29].  In this study, the scale has demonstrated good reliability with Cronbach’s α of 0.86.

Comment 5: Limitations of cross-sectional design. 

Response 5: Thanks for your feedback. The use of cross-sectional was addressed as a limitation for the study (P.10).

Comment 6: It is important to highlight and discuss the adjusted R2 value of the regression models. This value indicates the percentage of variance in sleep quality (or stress) that is explained by the other variables in the model. A low R2 would reduce the impact or ‘practical significance’ of the findings. 

Response 6: We did not use the linear regression analysis in this study.

Comment 7: Regression model: In the results section, when presenting the linear regression analysis, it must be confirmed that the model assumptions were met (such as normality of residuals and absence of multicollinearity).

Response 7: We did not use the linear regression analysis in this study.

Comment 8: Avoid inferences of causality: The text should be very careful when discussing the results. A cross-sectional study can only establish associations or correlations between TDPM, stress, and sleep quality.

Response 8: We appreciate your valuable comments.  We edit the discussion section and eliminate inference causality phrases (p. 8-10). 

Comment 9: Any language suggesting that ‘stress causes poor sleep quality’ or vice versa should be avoided. It should be changed to phrases such as ‘stress is significantly associated with...’.

Response 9: We appreciate your valuable comments.  We edit the discussion section (p.8-10). 

Comment 10: Recall bias: In the Limitations section, it is essential to add recall bias as a limitation, given that all data were collected through self-report (questionnaires) and participants must recall past symptoms and habits. 

Response 10: Thank you for your feedback. We highlighted the issue with recall bias in limitation section (p.10, Line 349-352). 

Comment 11: Relevance of the findings: The discussion is the ideal place to delve deeper into why the relationship found is important. For example, if TDPM is found to be a stronger predictor of poor sleep quality than perceived stress, this should be the main focus of the argument.

Response 11: Thank you for your feedback. We edit the discussion section and incorporated your feedback accordingly.

Comment 12: Recommendations in the conclusion and discussion should be more actionable. Instead of simply stating that more research is needed, the following suggestions could be made: Recommend longitudinal studies to confirm the direction of causality between stress and sleep. 

Response 12: Thank you for your feedback. We edited the conclusion section and integrated your comments (p.10). 

Comment 13: Suggest the implementation of specific stress management interventions (e.g., mindfulness or cognitive therapy) in the nursing curriculum to measure their impact on RMD symptoms and sleep quality

Response 13: Thank you for your feedback. We edited the discussion section and integrated your comments (p.8-10). 

Round 2

Reviewer 1 Report

Comments and Suggestions for Authors

The authors have significantly improved the manuscript. I would like to thank them for their detailed and point-by-point responses to my revision suggestions.

Reviewer 2 Report

Comments and Suggestions for Authors

The authors have addressed the recommendations made. This scientific article may be accepted.